# Seawater carbonate chemistry considerations for ocean alkalinity enhancement research: theory, measurements and calculations

Kai. G. Schulz[1], Lennart T. Bach[2], Andrew G. Dickson[3]

[1] Faculty of Science and Engineering, Southern Cross University, Lismore, NSW, Australia

[2] Institute for Marine and Antarctic Studies, University of Tasmania, Hobart, TAS, Australia

[3] University of California at San Diego, Scripps Institution of Oceanography, 9500 Gilman Drive, La Jolla, CA 92093, USA

*Correspondence to*: Kai. G. Schulz (kai.schulz@scu.edu.au)

**Abstract.** Ocean alkalinity enhancement (OAE) is a proposed marine carbon dioxide removal (mCDR) approach that has the potential for large-scale uptake of significant amounts of atmospheric carbon dioxide ($CO_2$). Removing anthropogenic legacy $CO_2$ will be required to stabilise global surface temperatures below the 1.5-2 °C Paris Agreement target of 2015. In this chapter we describe the impacts of various OAE feedstocks on seawater carbonate chemistry, as well as pitfalls that need to be avoided during sampling, storage and measurement of the four main carbonate chemistry parameters, i.e. dissolved inorganic carbon (DIC), total alkalinity (TA), pH and $CO_2$ fugacity ($fCO_2$). Finally, we also discuss considerations in regard to calculating carbonate chemistry speciation from two measured parameters. Key findings are 1) theoretical $CO_2$ uptake potential (global mean of 0.84 moles of $CO_2$ per mole of TA added) based on carbonate chemistry calculations are probably secondary in determining in which oceanic region OAE would be best, 2) carbonate chemistry sampling is recommended to involve gentle pressure filtration to remove calcium carbonate ($CaCO_3$) that might have been precipitated upon TA increase as otherwise interfering with a number of analyses, 3) samples for DIC and TA can be stabilised to avoid the risk of secondary $CaCO_3$ precipitation during sample storage, and 4) some OAE feedstocks require additional adjustments to carbonate chemistry speciation calculations using available programs and routines, for instance if seawater magnesium or calcium concentrations are modified.

## 1 Seawater carbonate chemistry revisited

In the following sections the basic concepts of the seawater carbonate system are laid out, together with definitions of the key parameters.

### 1.1 Acid-base parameters and equilibrium chemistry

The acid-base equilibrium chemistry of seawater has been described in the Guide to Best Practices for Ocean $CO_2$ measurements (Dickson et al. 2007), and the carbonate chemistry chapter in the Guide to Best Practices for Ocean Acidification Research and Data Reporting (Dickson 2010). These publications are open access and provide practical

guidance related to studying the carbon dioxide system in seawater. They describe tested approaches for measuring the four main parameters that are commonly used to constrain seawater acid-base chemistry and provide recommended values for the various equilibrium constants (as functions of salinity and temperature, and at 1 atm total pressure). Another key resource, although not open access, is the book 'CO$_2$ in seawater: Equilibrium, Isotopes, Kinetics' (Zeebe & Wolf-Gladrow 2001), which offers comprehensive theoretical knowledge on all aspects relevant for carbonate chemistry related research, including OAE.

The marine carbonate system can be constrained by knowing next to temperature, salinity and pressure, two carbonate chemistry parameters (Zeebe & Wolf-Gladrow 2001). Additional data such as silicic acid or phosphate concentrations might also be required, although these have typically limited influence on carbonate chemistry speciation. Four carbonate chemistry parameters are commonly measured (carbonate ion concentration measurements are less common). These are:

- DIC - the total dissolved inorganic carbon (expressed as an amount content of carbon atoms, i.e. moles per kg of seawater) in a sample of seawater (Eq. 1)
- TA - the total alkalinity (expressed as an amount content of hydrogen ions that are equivalent to the bases – defined relative to a specified equivalence point) in a sample of seawater (Eq. 2)
- pH$_T$ - a measure of the amount content of *total* hydrogen ions in a sample of seawater (Eq. 3)
- $f$CO$_2$: the fugacity of carbon dioxide in air that is in solubility equilibrium with a sample of seawater (Eq. 4)

(**Eq. 1**)  $DIC = [CO_2^*] + [HCO_3^-] + [CO_3^{2-}]$

with $[CO_2^*]$ denoting the sum of dissolved $[CO_2]$ and $[H_2CO_3]$.

(**Eq. 2**)  $TA = [HCO_3^-] + 2[CO_3^{2-}] + [B(OH)_4^-)] + [OH^-] + [HPO_4^{2-}] + 2[PO_4^{3-}] + [H_3SiO_4^-] + [NH_3] + [HS^-]$
$$- [H^+]_F - [HSO_4^-] - [HF] - [H_3PO_4] + \ldots - \ldots$$

with the last two terms denoting minor contributions of additional proton acceptors and donors.

(**Eq. 3**)  $pH_T = [H^+]_F + [HSO_4^-]$

with $[H^+]_F$ describing the free hydrogen ion concentration.

(**Eq. 4**)  $fCO_2 = K_o[CO_2^*]$

with $K_0$ denoting the temperature and salinity dependent carbon dioxide solubility, or Henry's law constant. Another important carbonate chemistry concept is that of the calcium carbonate (CaCO$_3$) saturation state (Eq. 5).

(**Eq. 5**)  $\Omega = \dfrac{[Ca^{2+}][CO_3^{2-}]}{K_{sp}}$

with $K_{sp}$ describing the temperature, salinity and pressure dependent solubility product of a particular calcium carbonate morphotype such as calcite or aragonite, and $[Ca^{2+}]$ and $[CO_3^{2-}]$ being respective in-situ concentrations. When $\Omega$ is above 1 CaCO$_3$ precipitation is thermodynamically favoured, while below 1 it is its dissolution.

The equilibrium constants for the amount contents of the various constituents of the acid-base equilibria in a sample of seawater are essentially dependent on the major ion composition of the seawater (usually approximated by the salinity), the temperature of the seawater, and the total pressure, i.e. water depth. Hence if salinity, temperature or pressure of a seawater sample change, the various equilibrium constants will necessarily change. Thus, the $pH_T$ and $fCO_2$ of the seawater will change as a consequence, and it is important to note the salinity, temperature and pressure at which the measurements were made. For TA and DIC this is not a problem. The individual amount contents of unionized dissolved carbon dioxide, bicarbonate ion, and carbonate ion each change as salinity, temperature and/or pressure change, but their total sum does not (the amount of carbon atoms that are in an inorganic form is conserved). Of course, this assumes that any redox reactions involving carbon do not occur on the relatively short timescales appropriate to the changes of salinity, temperature or pressure. The same applies to the sum of the individual TA components.

Any two of the five measurable parameters (DIC, TA, $pH_T$, $fCO_2$ and $CO_3^{2-}$) can – in principle – be used together with salinity, temperature, and pressure (defining suitable equilibrium constants) to calculate the state of the carbon dioxide acid-base system in a seawater sample. The parenthetic note: "in principle", is an acknowledgement that – as a consequence of errors in the measurements of these parameters and in the measurement of the associated equilibrium constants – the results from using alternate pairs of parameters will likely not be identical (Orr *et al.*, 2018). When total alkalinity is used as a measured parameter, it implicitly incorporates all other acid-base equilibria occurring within the sample of seawater, *i.e.* in addition to the inorganic $CO_2$ system (Eq. 2). This is relatively straightforward if the total amount content of each acid-base system present has been measured directly, can be calculated from sample salinity, or can safely be neglected. For each acid-base system considered, the acid-dissociation constants also need to be known. A useful consequence of the TA definition is that TA does not change when $CO_2$ is in- or outgassing. However, there is a growing awareness that the presence of organic acids and bases in seawater can complicate the use of total alkalinity as a measurable parameter (see *e.g.*, Fong & Dickson, 2019). The contribution of such species to total alkalinity is usually neglected, especially for open ocean seawater samples.

**1.2 Deffeyes diagrams**

Deffeyes (1965) published a detailed description of how to use contour diagrams plotting individual seawater $CO_2$ system properties as a function of total alkalinity and total dissolved inorganic carbon for a seawater of a specified salinity and at a particular temperature (and pressure). Insofar as this chapter is aimed at studies of ocean alkalinity enhancement, we feel such diagrams may be useful to help visualising the compositional changes that are involved in such processes.

When an alkaline agent such as sodium hydroxide (NaOH) is used to increase total alkalinity, such addition does not change total dissolved inorganic carbon (the vertical red line, Fig. 1). Alternatively, alkalinity could be added as a soluble inorganic carbonate salt (*e.g.* ikaite, $CaCO_3.6H_2O$, or $Na_2CO_3$). Both would increase TA and DIC in a ratio 2:1 (blue line, Fig. 1). As that modified water sample then equilibrates with the atmosphere, *e.g.*, takes up $CO_2$ from the air, DIC increases. It is

straightforward to assess the potential of 1 kg of an alkalinity-enhanced seawater to remove $CO_2$ from the atmosphere. It is equivalent to the change in DIC after re-equilibration to the initial $fCO_2$ prior to the alkalinity addition (horizontally moving from the tip of the alkaline agent arrow to the initial $fCO_2$). This concept takes into account that marine Carbon Dioxide
Removal (mCDR) also includes scenarios in which an oceanic $CO_2$ source is reduced. Similar diagrams can be drawn to assess the impact of various TA additions on $\Omega$ or pH (Fig. 1b,c).

### 1.3 OAE impacts on seawater and potential secondary $CaCO_3$ precipitation

As mentioned above, ocean alkalinity can be increased by various means (see also Eisaman et al. 2023, this Guide). First,
there are multiple minerals found in natural rocks which release TA upon dissolution in seawater. Dissolution kinetics depend on how finely the minerals have been milled as well as on the mineral itself (e.g. Anbeek 1992). For instance, olivine, a magnesium/iron silicate - $(Mg,Fe)_2SiO_4$, dissolves relatively slowly (for the purpose of OAE), on the order of weeks to months and appears to have a relatively low solubility (Monserrat et al. 2017, Flipkens et al. 2023). On the other side of the spectrum, brucite, $Mg(OH)_2$, dissolution rates can be on the order of hours  (Moras et al. 2023a). Another way to
increase oceanic TA is to add it in 'liquid' or aqueous form, i.e. hydroxide ions – $OH^-$, which can be produced electrochemically from seawater (e.g. Rau et al. 2013, but also see Eisaman et al, 2023, this Guide).

Whichever the approach, once alkalinity is increased it raises pH, shifting carbonate chemistry speciation towards lower aqueous carbon dioxide ($[CO_2]$) and higher carbonate ion concentrations ($[CO_3^{2-}]$), as well as saturation states for various
calcium carbonate ($CaCO_3$) morphotypes such as calcite ($\Omega_{calc}$) and aragonite ($\Omega_{arag}$). In this context, saturation states can be considered critical thresholds, and depending on application there are levels that should not be exceeded for extended periods of times. Avoiding critical saturation thresholds is important because $CaCO_3$ will precipitate inorganically through a number of mechanisms at higher $\Omega$ values. Such secondary precipitation should be avoided as it reduces the oceanic uptake capacity for atmospheric $CO_2$ and can even lead to runaway $CaCO_3$ precipitation where more TA is removed than had been added,
reducing typical $CO_2$ uptake potential by a factor of eight (Fuhr et al. 2022, Moras et al. 2022).

In seawater there are three mechanisms for inorganic $CaCO_3$ precipitation, 1) homogeneously - in the absence of any soluble or particulate surfaces, 2) pseudo-homogeneously – in the presence of particulate or colloidal organics, and 3) heterogeneously – in the presence of solid mineral phases (Marion et al. 2009). The critical $\Omega$ threshold beyond which
$CaCO_3$ formation would occur is highest for homogeneous and lowest for heterogeneous precipitation, with pseudo-homogeneous in between (Morse et al. 2007). In a natural setting, homogeneous precipitation is unlikely, as seawater is hardly void of organic or inorganic particles and/or colloids. For pseudo-homogeneous $CaCO_3$ precipitation, the critical $\Omega_{arag}$ threshold is about 12.3 at a salinity 35 and 20°C (Marion et al. 2009). For heterogeneous precipitation, it depends on the so-called lattice compatibility between $CaCO_3$ and the mineral surfaces it precipitates onto. For instance, the mineral $CaCO_3$ has
a perfect lattice compatibility, meaning that any existing pre-cursors will lead to precipitation at $\Omega$ above 1 (Zhong&Mucci

1989). The rate of precipitation is exponentially increasing with $\Omega$, about 5-fold for a doubling in saturation state. In contrast, other mineral surfaces have lower lattice compatibilities, for instance $CaCO_3$ precipitation was observed for CaO and $Ca(OH)_2$ above an $\Omega$ of 7, while for $Mg(OH)_2$ rates appeared to be even further reduced (Moras et al. 2022, 2023a). Critical $\Omega$ thresholds are also influenced by grain size, salinity and dissolved organic matter concentrations (Simkiss 1964, Chave & Suess 1970, Moras et al. 2023a), and most likely temperature as well.

The risk of secondary precipitation is particularly high when TA has been increased but atmospheric $CO_2$ has not yet entered the ocean, i.e. $\Omega$ is still high. However, there is also the possibility to equilibrate seawater with respect to atmospheric $CO_2$ during TA addition which would reduce the possibility of secondary $CaCO_3$ precipitation as $\Omega$ would be much lower for the same amount of TA added (Bach et al. 2019). Likewise, dilution of TA-enhanced with unperturbed seawater has been shown to effectively slow/stop secondary precipitation due to a reduction in $\Omega$ (Moras et al. 2022).

### 1.4 TA additions and critical $\Omega$ thresholds in the surface ocean

From a practical OAE standpoint, there will always be an $\Omega$ that one will try not to go beyond for extended periods of time to minimise potential secondary $CaCO_3$ precipitation. How much TA can be added depends therefore on initial in-situ seawater $\Omega$. Globally, both surface ocean $\Omega_{arag}$ and $\Omega_{calc}$ are highly correlated, at least on larger scales, with temperature and salinity. On smaller scales, there is also a biological component, i.e. photosynthesis increasing $\Omega$ and respiration decreasing it. Impacts of temperature and salinity changes on carbonate chemistry speciation are two-fold. While there is a direct effect of temperature on carbonate chemistry speciation, the bigger impact on $\Omega$ stems from the higher $CO_2$ solubility in low temperature waters. The resulting higher $[CO_2]$ and lower TA/DIC at atmospheric equilibrium, leads to a shift towards lower $[CO_3^{2-}]$, and hence $\Omega$. This is because $[CO_3^{2-}]$ is approximately equal to the difference between carbonate alkalinity (CA) and DIC in the ocean (Schulz & Maher 2023). This largely explains the latitudinal gradient of surface ocean $\Omega$ (Jiang et al., 2015). Additionally, the temperature sensibility of the apparent calcium carbonate solubility product ($K_{sp}$') also contributes to the latitudinal $\Omega$ gradient (Mucci, 1983).

When it comes to salinity, there is also a direct effect but the major driver is that in oceanic waters DIC and TA scale with salinity (mostly driven by water evaporation and precipitation), meaning that they will be significantly reduced at lower salinities. And for a similar oceanic pH at equilibrium with atmospheric $CO_2$, reducing salinity and DIC, for instance 5-fold, will reduce $[CO_3^{2-}]$ by a similar amount. In turn, this will reduce $\Omega$, together with the salinity-related reductions in $[Ca^{2+}]$ at lower salinities. Globally, sea surface temperatures are highest in the tropics and subtropics and decrease towards higher latitudes (Figure 2A). Similarly, there is a tendency towards higher salinity at low latitudes and lower salinities at high latitudes (Figure 2B). In concert, this leads to increasing $\Omega$ from polar to tropical waters. For instance, while there are polar regions in which $\Omega_{arag}$ is close to 1 on an annual basis, in large parts of the tropical surface ocean $\Omega_{arag}$ can be close to 4

(Figure 2C). Finally, as mentioned earlier, there are also biological drivers of sea surface $\Omega$, for which a good indicator is $CO_2$ fugacity. Higher than current atmospheric pressures of ~425 µatm are indicative of respiratory $CO_2$ generation, reducing $\Omega$, while pressures below are typically driven by photosynthetic $CO_2$ fixation, increasing $\Omega$. This modulating effect of biology on temperature- and salinity-driven global surface ocean $\Omega$ distributions can be seen by the upwelling of $CO_2$-enriched deep waters in the tropical Pacific where elevated $fCO_2$ results in lower $\Omega_{arag}$ (Figure 2C, D). Altogether, this

determines how much TA can be added before a critical $\Omega$ threshold is reached. Assuming that $\Omega$ should not surpass a threshold of 5 to avoid secondary $CaCO_3$ precipitation (Moras et al. 2022), 5 times higher TA additions would be possible at high in comparison to low latitudes (Figure 2E).

The amount of atmospheric $CO_2$ that can be taken up for a certain TA increase, depends on the uptake factor $\eta_{CO2}$,

(Humphreys et al. 2018, Tyka et al. 2022). It denotes the molar ratio of the concomitant DIC increase relative to the amount of TA added upon establishing initial seawater $fCO_2$ by air-sea gas exchange. It is mainly determined by sea surface temperatures, with minor effects of salinity and biology. It ranges between 0.77 and 0.96, with higher values at lower temperatures closer to the poles (Figure 2F). The resulting DIC uptake potential for global surface waters, for a critical $\Omega$ threshold of 5, ranges between only 50 to up to 400 µmol kg$^{-1}$, being higher at high and lower at low latitudes (Figure 1G). It

is noted though, that the $CO_2$ uptake factor plays only a minor role, and most differences in regional $CO_2$ uptake potential are driven by the amount of TA that can be added before a critical $\Omega$ threshold is reached and/or the amount of time a water parcel stays in contact with the atmosphere (see next section). The uptake factor can be empirically estimated using available carbonate chemistry calculation tools or calculated analytically, for which there are also dedicated routines available (Humphreys et al. 2018)

**1.5 Cautionary note on apparent trends in regional differences for atmospheric $CO_2$ uptake**

When looking at the distribution of global DIC uptake potential calculated for a specific OAE application ($\Omega$ threshold), there appears to be a general trend for higher uptake at high and lower at low latitudes. However, that only applies to the open ocean, as coastal areas are not included in the underlying GLODAP climatology. And there, the situation is likely to be

quite different, as these areas are heavily impacted by terrestrial influences such as freshwater input but also anthropogenic disturbances such as nutrient runoff which affects biological processes, and hence the $CO_2$ uptake potential (see above), in particular as biological activity can be orders of magnitude higher than in the open ocean. Additionally, seasonal changes to carbonate chemistry speciation, which can be quite large, are also not captured.

Furthermore, it is only an uptake potential and the realised increase in DIC critically depends on regional gas transfer velocities, i.e. how quickly atmospheric $CO_2$ is equilibrating with the surface ocean, and for how long the surface water with increased TA is in contact with the atmosphere before eventually being subducted. There are regions with relatively quick $CO_2$ equilibration and surface waters remaining in contact with the atmosphere on the relevant timescales, meaning that

about a year after the TA increase ~80-100% of the $CO_2$ uptake potential could be realised (He & Tyka 2023). In contrast, there are regions with either quite slow equilibration and/or where significant portions of TA enriched surface waters lose contact with the atmosphere for relatively long times, e.g. regions of deep water formation. Here, the realised $CO_2$ uptake potential might only be 50% or less after a couple of decades (He & Tyka 2023).

Finally, a water mass that has received a TA addition might change its physical properties such as temperature, for instance when moving from high to low latitudes. As such $\eta_{CO2}$ decreases along the trajectory, meaning that the $CO_2$ uptake potential declines. Hence, instead of using $\eta_{CO2}$ at the site of TA addition to estimate its DIC uptake potential, it appears more appropriate to use a global mean of 0.84.

Last but not least, it is advisable to not treat $\eta_{CO2}$ or the DIC uptake potential as a measure of OAE efficiency. The latter should be assessed in two steps, i.e. first how much of a stable TA increase can be achieved, accounting for potential $CaCO_3$ precipitation, as well as potentially shifting natural TA source/sink terms, for instance in sediments (Bach 2023). And secondly, how much of DIC is actually then taken up given a particular air-sea gas exchange and exposure time. Such approach takes into account the quite different time scales of TA and DIC increases.

## 2 Sampling and storage of TA-enriched water

Sampling of discrete DIC, TA, pH and $fCO_2$ should generally follow protocols described in Standard Operating Procedure (SOP) 1 by Dickson et al. (2007). One caveat, however, is that these protocols were developed for common oceanic carbonate chemistry conditions. Here, we discuss additional measures for sampling and sample storage that may be necessary for carbonate chemistry conditions specific to OAE (i.e. high TA, high pH, and high $\Omega$). We acknowledge that many of the procedures described below are based on anecdotal evidence and need to be further scrutinized with increasing maturation of OAE research and ongoing improvements of this guide.

### 2.1 Specific problems to consider

In general, there are 4 processes that can alter carbonate chemistry conditions in a sample during its collection and storage. These are:

1) modifications via air-sample gas exchange (sampling)
2) modifications via biological activity such as respiration (storage)
3) modifications via the precipitation of $CaCO_3$ (storage)
4) modifications via diffusion of $CO_2$ into or out of sampling container walls (storage)

These four processes affect carbonate chemistry parameters differently and the effect size also depends to a large degree on the sample itself (Table 1). Sampling and storage procedures that describe how to deal with processes 1 and 2 can be found in SOP 1 by Dickson et al. (2007), such as how samples should be collected and stored, including thorough instructions of what materials are recommended to be used and how they should be prepared.

CaCO$_3$ precipitation (process 3) has received less attention in SOP 1. This is because CaCO$_3$ precipitation is very slow under most natural seawater carbonate chemistry conditions and is therefore not considered to affect carbonate chemistry conditions during sampling and storage of such waters. However, CaCO$_3$ precipitation accelerates exponentially the further $\Omega$ deviates from 1 (Zhong & Mucci 1989). Extreme $\Omega$ values which are much higher than what is commonly observed in the oceans will be frequently encountered in OAE research. Indeed, Subhas et al. (2022) experienced this problem in their OAE experiments where TA and DIC concentrations in the OAE treatment samples declined during sample storage due to CaCO$_3$ precipitation. The precipitation of CaCO$_3$ reduces TA and DIC in a 2:1 molar ratio, also changing carbonate chemistry speciation (Figure 3). As discussed above, precipitation rates critically depend on $\Omega$ but are also influenced by the presence of organic/inorganic particles as precipitation nuclei (Zhang and Mucci, 1989; Marion et al. 2009, Moras et al., 2022; Fuhr et al., 2022), the Mg concentration (Moras et al. 2023a), or the concentration of dissolved organic carbon - DOC (e.g. Chave&Suess 1970, Pan et al. 2021, Moras et al. 2023a). Since, precipitation is usually difficult to predict in terms of how long a sample might be stable, it is advisable to treat the sample in such a way that the potential problem of precipitation is minimised/mitigated.

In the following we describe procedures to avoid changes in carbonate chemistry conditions upon sampling and storage of DIC, TA, pH, and $f$CO$_2$.

## 2.2 General sampling considerations: filtration

It is advisable to filter carbonate chemistry samples upon their collection to remove particles. In oceanography, the cut-off between what is considered to be dissolved or particulate has been operationally defined by passing or being retained on a GF/F filter. Such a filter has a nominal pore size of 0.7 μm, but smaller pore sizes, for instance 0.2 μm which would remove bacteria at the same time and theoretically produce a sterile sample, have also worked in the past. Filtration has multiple benefits and might even be required:

1. Particles can serve as precipitation nuclei, which catalyse CaCO$_3$ precipitation during storage. Removing particles via filtration will therefore help to reduce the risks of secondary precipitation.
2. Removing particles will be necessary in case CaCO$_3$ precipitation has already occurred in-situ to not interfere with the measurements. This is because typical analytical procedures to measure DIC and TA rely on acidifying the seawater sample during analysis, meaning that any CaCO$_3$ present would re-dissolve and be measured as additional DIC and TA.

3. Filtration of the sample is also useful to remove substrate for biological breakdown and organisms that respire organic carbon. Respiration has limited influence on TA but strongly affects DIC, pH, $f\text{CO}_2$ and $[\text{CO}_3^{2-}]$ (Table 1). However, for samples intended to be stored longer than a few days, sterile filtration is not recommended but fixation with mercuric chloride ($\text{HgCl}_2$; Dickson et al. 2007).

Filtration can have negative side-effects that need to be considered and mitigated. Most obviously, filtration requires additional sampling gear which introduces contamination risks. Thus, all filtration gear must be prepared appropriately following SOP 1 in Dickson et al. (2007). The most critical issue is that filtration increases the risk of air/sample $\text{CO}_2$ gas exchange. This risk can be reduced when the sample is handled by gentle pressure-filtration (as opposed to vacuum filtration which is prone to gas exchange, i.e. $\text{CO}_2$ degassing) in a closed system. For example, the sample could be pumped via a

peristaltic pump using Tygon tubing (relatively impermeable for $\text{CO}_2$) through a membrane filter directly into the sampling bottle, filling it from bottom to top with significant overflow (Dickson et al. 2007, Schulz et al. 2017). Alternatively, the sample could be filled into a syringe (again, potential air/water gas exchange has to be minimised) and filtered through a syringe filter, again from bottom to top with overflow. $\text{CO}_2$ gas exchange would change DIC and hence is also a problem for pH, $f\text{CO}_2$ and $[\text{CO}_3^{2-}]$ samples, which would also change. While TA samples are not affected by air/sample $\text{CO}_2$ gas

exchange, they are sensitive to evaporation, so prolonged exposure to the atmosphere must also be avoided.

## 2.3 General storage considerations

Samples for DIC, TA, pH, $f\text{CO}_2$ and $[\text{CO}_3^{2-}]$ should be prepared for storage and stored as described in SOP 1 in Dickson et al. (2007). If these samples have relatively high $\Omega$ values (as a rule of thumb $\Omega_{\text{arag}} > 12$ for filtered and $\Omega_{\text{arag}} > 5$ for

unfiltered samples), they most likely require additional measures to avoid changes through $\text{CaCO}_3$ precipitation that could eventually occur during storage.

We reiterate that much of what is described below, while appearing to be logical, remains to be tested and validated in dedicated laboratory studies. Thus, care must be taken when adopting these preliminary recommendations.


## 2.3.1 Preparing DIC samples for storage

DIC changes with $\text{CaCO}_3$ precipitation in a 1:1 molar ratio (Figure 3). Eventual $\text{CaCO}_3$ precipitation can be avoided by adding a strong acid (e.g. hydrochloric acid - HCl) to the sample, although precautions have to be taken (see below). HCl reduces TA and hence $\Omega$. Such addition also decreases pH, and increases $f\text{CO}_2$ in the sample but leaves DIC concentrations

untouched. Increasing sample $f\text{CO}_2$ might, however, be a problem, if acid is added in excess to the added TA, $f\text{CO}_2$ will eventually rise above atmospheric levels and increase the risk of outgassing which would decrease DIC.

Hence, to mitigate $\text{CO}_2$ outgassing it is advisable to dose the HCl additions to just compensate the TA addition, bringing $\Omega$ down to typical seawater levels, i.e. $\Omega_{\text{arag}} \sim 2$. Also, prior to full mixing, there might be localised high $f\text{CO}_2$

microenvironments, hence exposure of the sample to air after acidification should be avoided, for instance by closing the lid. Again, equilibrated OAE samples most likely do not require such an acidification step (see 2.3.1).

To reduce sample dilution, it is recommended to use strong acids (e.g. HCl) at relatively high concentrations. The dilution of the DIC sample with HCl will need to be considered to calculate the DIC concentration of the undiluted sample.


### 2.3.2 Preparing TA samples for storage

TA changes due to $CaCO_3$ precipitation occur in a 2:1 molar ratio (Fig. 3A). $CaCO_3$ precipitation, due to initially high $\Omega$, can be avoided by briefly bubbling the TA sample with pure $CO_2$ gas. $CO_2$ decreases $\Omega$ and pH, and increases DIC and $f\mathrm{CO_2}$ but
leaves TA unaffected (note that $CO_2$ is not part of the TA definition - Eq. 2). The approach was tested with 120 mL of seawater sample where high purity $CO_2$ gas (4.5, i.e. 99.995%) was bubbled via Tygon tubing through a clean 1 mL pipette tip into the sample for 10 seconds (Lenc et al. 2023). The $CO_2$ flow rate was not determined but adjusted so that the bubbling did not lead to spill over in the 125 mL sample bottle. Measured pH declined from initially ~8.7 to well below 6 within 10 seconds so that $\Omega_{arag}$ went from ~9.5 to much below 1. Under these conditions, TA loss through $CaCO_3$ precipitation is
impossible. It is noted that bringing the pH down to pH levels prior to the TA addition, i.e. around 8, would have reduced $\Omega_{arag}$ sufficiently ($< 5$) to avoid $CaCO_3$ precipitation during sample storage if the sample had been filtered.

However, there are several aspects to consider when employing this sample fixation approach. First, impurities in the $CO_2$ gas could contaminate the TA sample. Particle impurities can be avoided by using filters. Gas impurities (e.g. $NH_3$) will be
more difficult to remove so that it is important to use high-purity gases and test for potential TA impurities. Second, bubbling with dry $CO_2$ gas directly leads to evaporation of water, increasing TA. It is unclear at this stage how relevant this problem is for a short bubble burst but it could be avoided by saturating the $CO_2$ gas stream with $H_2O$ before bubbling the sample as for example described in Moras et al. (2023b).

Finally, atmosphere-equilibrated OAE samples which have $\Omega$ levels below the critical thresholds described above, should not require $CO_2$ bubbling. Although it is advised to actually calculate expected sample $\Omega$ from estimates of TA and DIC prior to storage.

### 2.3.3 pH and $f\mathrm{CO_2}$ samples

Samples for pH and $f\mathrm{CO_2}$ cannot be stabilised with any of the methods described above. Both, the addition of HCl or $CO_2$ would alter the carbonate chemistry speciation, including pH and $f\mathrm{CO_2}$. There may be methods to reduce $CaCO_3$ precipitation via the addition of certain inhibitors such as DOC, but such methods would need to be developed. Thus, the recommendation for pH and $f\mathrm{CO_2}$ of samples with high $\Omega$ is to measure them immediately after collection.

### 3. Carbonate chemistry measurements of TA-enriched water

Concerning measurements of carbonate chemistry parameters, there are different needs depending on scientific research questions. Two levels of uncertainty have been proposed, one being able to detect relatively small carbonate chemistry speciation changes of 'climate' signals, and the other being able to monitor shorter term or spatial variability which can be considerably larger, termed 'weather' signals (Newton et al. 2015).

Being able to resolve 'climate' signals requires measurement uncertainties equal or better than 2 $\mu$mol kg$^{-1}$ for DIC and TA, 0.003 for pH and 0.5% for $CO_2$ fugacity. In comparison, for 'weather' signals better than 10 $\mu$mol kg$^{-1}$ for DIC and TA, 0.02 for pH and 2.5% for $f$CO$_2$ are sufficient (Newton et al. 2015).

Concerning OAE, both thresholds could be aimed for depending on application. For example, if it is to monitor initial TA changes upon an alkalinity addition which can be on the order of several hundreds of $\mu$mol kg$^{-1}$ of seawater, then aiming to resolve such 'weather' signal with an uncertainty of better than 10 $\mu$mol kg$^{-1}$ is probably sufficient. However, once this signal gets diluted and to monitor $CO_2$ ingassing and DIC increase over timescales of months an uncertainty for 'climate' signals must be aimed for. In fact, depending on dilution, this might even not be enough, highlighting the fact that monitoring, reporting and verification (MRV), i.e. determining how many carbon credits can be assigned to a certain OAE effort in the end, is likely to require modelling (Fennel et al. 2023, Ho et al. 2023, this Guide).

In any case, a first step before measuring any carbonate chemistry sample would be to assess the accuracy (how far off are measurements with respect to certified reference materials) and precision (what is the standard deviation of replicate measurements) of one's instrumentation, allowing to estimate the overall uncertainty for each parameter in question. Another important point to consider is that OAE samples can be far off the concentrations of typical certified reference material, meaning that checking for linearity can be important (see below for details).

Finally, for any measurements described below, the starting point should be the Guide to Best Practices of ocean $CO_2$ measurements (Dickson et al. 2007). Hence, with the exception for $f$CO$_2$, below we are mostly referring to discrete, not in-situ, measurements, although the following recommendations should also be considered for the latter.

### 3.1 Measuring DIC

Acknowledging that OAE samples for DIC can be prone to air/sample gas exchange (see section 2.2.3.1), if the right precautions are taken during sampling and storage, they should be straightforward to measure. This is because DIC concentrations in a 'non-equilibrated' OAE sample will be the same as for untreated seawater samples. However, in samples

equilibrated with the atmosphere DIC can be several hundreds of micromoles higher than typical seawater samples or certified reference material, meaning that the linearity of the measurement instrument and procedure has to be ascertained.

Linearity can be checked for by preparing and measuring $Na_2CO_3$ (ultrapure and dried at 280 °C for 2 h) solutions of increasing DIC in Milli-Q water (ideally prepared from a concentrated stock solution), covering the required concentration range, and comparing the fractional offset of measured against theoretical concentrations for each measurement point (Figure 4). If the fraction does not change with concentration, the system response is linear and a 'one-point calibration' against a certified reference material will be sufficient. Using $NaHCO_3$ to create a DIC gradient is not ideal as it is not available in

ultrapure form and without further modifications it would create high $fCO_2$ samples which would need additional precautions because of potential $CO_2$ degassing (note that $\Delta fCO_2$ is much higher for $NaHCO_3$ than for $Na_2CO_3$ samples).

Finally, it is again emphasised that sample filtration prior to measurement is most likely a crucial step to first stabilise the sample in terms of reducing the potential for $CaCO_3$ precipitation and second, to remove any $CaCO_3$ that would have precipitated prior to sampling as it would interfere with the DIC measurement (see section 2.2.2 for details).


### 3.2 Measuring TA

When it comes to measuring TA, samples are likely to be enriched in comparison to the typical surface ocean. Hence, checking for linearity of the instrument setup like in the case for DIC is important. Again, this can be done by preparing suitable calibration standards, covering the required concentration range. However, unlike for relatively straight forward DIC

measurements this requires interpreting pH titration data on the basis of chemical acid/base equilibria in a well-defined ion matrix, called seawater (precisely defining the zero level of protons). Hence, simply using $Na_2CO_3$ or $NaHCO_3$ in Milli-Q water is not the ideal option. Instead, to cover the higher TA range, suitable amounts of $NaHCO_3$ could be added to seawater ($Na_2CO_3$ is not advisable as it is quite alkaline and might induce some sort of carbonate precipitation, impacting TA). If the TA range also requires concentrations lower than typical seawater, for instance for samples that had seen substantial amounts

of $CaCO_3$ precipitation, then this can be easily achieved by diluting natural seawater with Milli-Q water (weight by weight) and taking the change in salinity into account. Again, checking for changes in the ratio of measured to theoretical TA along the measured TA gradient will tell if the measurement system is linear, like for DIC (compare Figure 4).

A last thing to consider is that high TA samples can need considerably more titrant if the molarity of the acid is not

increased. Under certain circumstances this can lead to non-linearities, e.g. if the dosing unit needs calibration. Finally, as for DIC it is important that no $CaCO_3$ is present in the sample, hence an additional filtration step during sampling might be required.

### 3.3 Measuring pH

If pH is measured by a glass electrode, following the recommendations in Dickson et al. (2007), there should be no additional precautions required for OAE samples, whether they be of high TA and pH or high TA and DIC, hence typical seawater pH. Also, any $CaCO_3$ that has precipitated prior to sampling will not interfere with the measurements (potentiometric pH only). However, if $CaCO_3$ has precipitated post-sampling, this will have decreased the pH and an erroneous reading will be made. Concerning accuracy, potentiometric pH measurements can be less accurate than

spectrophotometric pH measurements (Bockmon & Dickson 2015).

For spectrophotometric measurements, one key element is the working range of the pH dye being used. It has been suggested that for sulfonephthalein indicator dyes it is between one pH unit below and above the indicator's $pK_2$, (see Hudson-Heck et al. (2021) and references therein). The latter is the pH for which the concentration of the double-unprotonated form of the

dye is equal to that of the single-protonated one (Byrne et al. 1988). The two most commonly used pH dyes for seawater are meta-cresol purple (mCP) and thymol blue (TB), with $pK_2$ of ~8.0 and ~8.5, respectively, at a temperature of 25°C and a salinity of 35, although there are a number of studies that have extended on the salinity and temperature range (see Hudson-Heck et al. (2021) and references therein). That would suggest that seawater pH ranging from 7 to 9.5 could be measured at high accuracy and precision. However, in practice accuracy can be influenced by dye impurities, and their effect can even be

dependent on pH. Hence, when unpurified dyes are used, it is highly recommended to check for the proper working range.

Indeed, we have found the working range to be significantly reduced for an unpurified mCP batch, for which we compared measured pH with calculated pH from measured DIC and TA (Figure 5). And while a pH deviation by 0.03 pH units at a pH of 8.5 might not seem much, it corresponds to about an offset of 30 $\mu$mol kg$^{-1}$ in DIC or TA if this pH measurement would

be used for carbonate chemistry calculations without correction, which is clearly way above both the 'weather' and 'climate' thresholds.

Another option for applying corrections is by direct means without the need for ancillary TA and DIC measurements (although they are recommended to check whether corrections are actually working within the desired pH range, which is

actually quite important as we will see below). For that purpose, the absorbance ratios obtained with an unpurified dye need to be corrected. While the actual procedure is beyond the scope of this chapter, the reader is referred to detailed instructions in Douglas & Byrne (2017) for mCP and Hudson-Heck et al. (2021) for TB. Again, for the uncorrected pH measurements we found a considerable offset of up to 0.06 pH units (Figure 6), corresponding to a ~60 $\mu$mol kg$^{-1}$ inaccuracy in TA or DIC if this pH measurement would be used for carbonate chemistry calculations. Furthermore, the pH error is smallest around $pK_2$

and increases below and above. While the impurity correction does fix the error close to $pK_2$, the observed trend remains. The fact that there is some sort of optimum curve behaviour around $pK_2$ points to an issue with the dye, rather than with measurements of DIC and TA. Hence, two separate linear corrections similar to the one with mCP (Figure 4) could be applied, or purified dyes being sourced.

Finally, as for potentiometric pH measurements the starting point for setting up spectrophotometric pH should be Dickson et al. (2007).

## 3.4 Measuring $f$CO$_2$

For $f$CO$_2$ measurements, air is being equilibrated with seawater, either via a CO$_2$ permeable membrane or in a so-called
showerhead equilibrator. As for potentiometric pH, any CaCO$_3$ in suspension will not affect $f$CO$_2$ measurements if precipitated pre-sampling. The only difference to typical seawater measurements is that $f$CO$_2$ can be relatively low for 'un-equilibrated' OAE, e.g. below 100 $\mu$atm for a TA addition of ~500 $\mu$mol kg$^{-1}$. Hence, full equilibration of seawater with the air to be measured might take a bit longer. Other than that, we do not see any particular issues, other than the generic problems with potential CaCO$_3$ precipitation if samples are stored (section 2.2.3).


## 4. Carbonate chemistry calculations

Carbonate chemistry calculations in an OAE context from two measured parameters can require additional considerations. One is that particular OAE applications not only change TA or eventually DIC but also the major ion composition of seawater. For instance, for calcium-based OAE, with for instance calcium oxide or hydroxide - CaO and Ca(OH)$_2$, for each
mol of TA half a mol of Ca$^{2+}$ will be added. This is the same for magnesium-based OAE, for instance with Mg(OH)$_2$ or olivine (forsterite). For example, increasing TA by 500 $\mu$mol kg$^{-1}$ would increase both calcium and magnesium ion concentrations by 250 $\mu$mol kg$^{-1}$, i.e. by 2.5% and 0.5% , respectively, over seawater background levels at a salinity of 35.

For calcium-based OAE this means that calculations of calcium carbonate saturation states have to factor in the increase in
the calcium to salinity ratio. In other words, using standard settings in various carbonate chemistry speciation calculation programs, e.g. CO2SYS, Seacarb, PyCO2SYS (for an overview see Orr et al. 2015), which derive calcium ion concentrations from salinity, will result in wrongly calculated saturation states. Hence, they would need to be calculated in a separate step from calculated carbonate ion and actual calcium concentrations (e.g. Moras et al. 2022), the programs being adapted, or special functions used (for instance available for Seacarb).


Furthermore, making changes to the matrix of the major ions in seawater also affects acid/base equilibria, i.e. stoichiometric dissociation constants such as for carbonic acid. Hence, one may also need to apply corrections to K$_1$ and K$_2$ for carbonic acid (Ben-Yakoov & Goldhaber 1972) which then can be used to calculate carbonate chemistry speciation, either by hand (Zeebe & Wolf-Gladrow 2001) or by adapting available programs. Furthermore, the solubility product of calcium carbonate,
K$_{sp}$, needs to be corrected which can be achieved by using the specific magnesium to calcium ratio in a seawater sample (Tyrell & Zeebe 2004). Finally, the effects of major (and minor) ion composition changes on carbonate chemistry speciation can also be assessed by Pitzer modelling (for an overview see Turner et al. 2016), and there have been several programs and

functions made available by the SCOR Working Group 145, MARCHEMSPEC: Chemical Speciation Modelling in Seawater to Meet 21st Century Needs (http://marchemspec.org/).

---

**Key recommendations**

Calcium carbonate ($CaCO_3$) will precipitate in seawater when the saturation state is beyond certain thresholds (critically depending on the presence of organic matter or mineral precursors, among others). Hence, when it comes to determine carbonate chemistry speciation for samples with significantly increased total alkalinity (TA) at regular dissolved inorganic carbon (DIC) concentrations, $CaCO_3$ that has precipitated prior to sampling has to be removed for most measurements (TA, DIC, spectrophotometric pH) as it interferes with the analytical procedures. Gentle pressure filtration, avoiding gas exchange of the sample with air, is recommended. Furthermore, if samples are to be stored prior to analysis, $CaCO_3$ precipitation needs to be prevented, which can be achieved for TA (lowering the calcium carbonate saturation state by brief sparging with $CO_2$), and DIC (acidifying the sample to reduce TA to natural pre-OAE conditions). Samples for $f CO_2$ and pH cannot be stabilised and should therefore not be stored. Any sample that has been stored should be checked visually for potential $CaCO_3$ precipitation on container walls or in suspension. For carbonate chemistry speciation calculations from two measured parameters, if experimental treatments have changed the $Mg^{2+}$ to $Ca^{2+}$ ratio in seawater this should be considered by applying corrections to stoichiometric equilibrium constants. Also, experimental changes to $Ca^{2+}$ concentrations mean that readily available calculation routines will report a wrong $CaCO_3$ saturation state as simply related to salinity. Again, corrections have to be made.

---


**Code/Data availability**

No original data has been produced for this manuscript and procedures to reproduce figures are described or referenced.

**Author contributions**

The manuscript was conceptually developed by all authors. Sections 1.1 and 1.2 were led by AGD, section 2 by LTB and the rest by KGS. All authors contributed to writing, graphing and editing.

**Competing interests**

Competing interests are declared in a summary for the entire volume at: https://sp.copernicus.org/articles/sp-bpoae-ci-summary.zip.

**Acknowledgements**

This is a contribution to the "Guide for Best Practices on Ocean Alkalinity Enhancement Research". We thank our funders the ClimateWorks Foundation and the Prince Albert II of Monaco Foundation. Thanks are also due to the Villefranche Oceanographic Laboratory for supporting the lead authors' meeting in January 2023. KGS and LTB also acknowledge

funding from the Carbon to Sea Initiative via the project OceanAlkAlign. LTB also acknowledges funding from the
Australian Research Council (FT200100846).

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

**Table 1:** Direction of change in a measured carbonate chemistry parameter due to various processes. Note that for process 1 the direction depends on whether the sample is over or undersaturated with respect to atmospheric $CO_2$. For process 4 the 620 direction depends on whether $CO_2$ is diffusing from the walls into the sample or vice versa. Also, the direction of TA change in process 2 depends on which form of inorganic nutrients are being released during organic matter remineralisation

|  | Air-sample $CO_2$ exchange (process 1) | Respiration/Remineralisation (process 2) | $CaCO_3$ precipitation (process 3) | Absorption and diffusion of $CO_2$ to/from walls (process 4) |
|---|---|---|---|---|
| TA | N/a | + / - | - | N/a |
| DIC | + / - | + | - | - / + |
| pH | - / + | - | - | + / - |
| $fCO_2$ | + / - | + | + | - / + |
| $[CO_3^{2-}]$ | - / + | - | - | + / - |





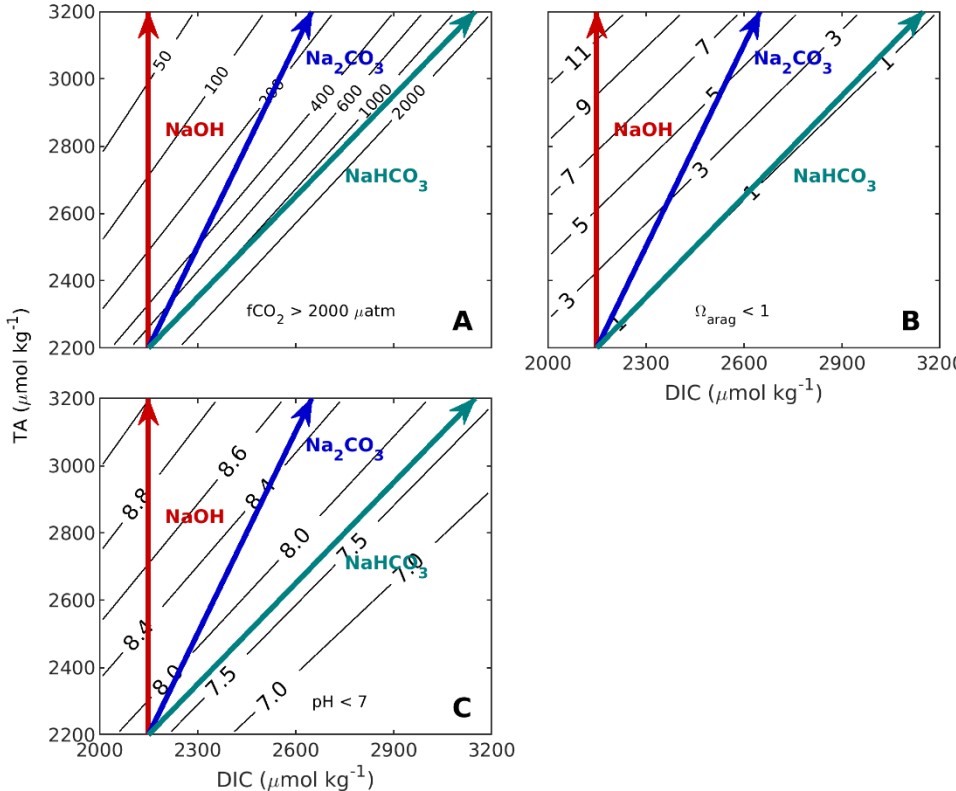

**Figure 1:** Implications of changes in $C_T$ and $A_T$ on the properties of undersaturated seawater appropriate to the California Current region. (S = 33.5, t = 10 °C, $A_T$ = 2200 µmol kg$^{-1}$, $C_T$ = 2150 µmol kg$^{-1}$). (A) f($CO_2$); (B) aragonite saturation state ($\Omega_{arag}$); (C) $pH_T$. Calculations were carried out with the Matlab version of CO2SYS (Pierrot et al., 2006) using the stoichiometric dissociation constants for carbonic acid from Sulpis et al. (2020), for sulphuric acid by Dickson et al. 1990, and total boron from Uppström (1974). The red line indicates the effect of adding NaOH, the blue line of adding $Na_2CO_3$ and the teal line of adding $NaHCO_3$. If a different initial seawater is chosen, this whole grouping (red, blue and teal lines) can be translated (moved without any distortion or rotation) so that its initial position is elsewhere on these figures.


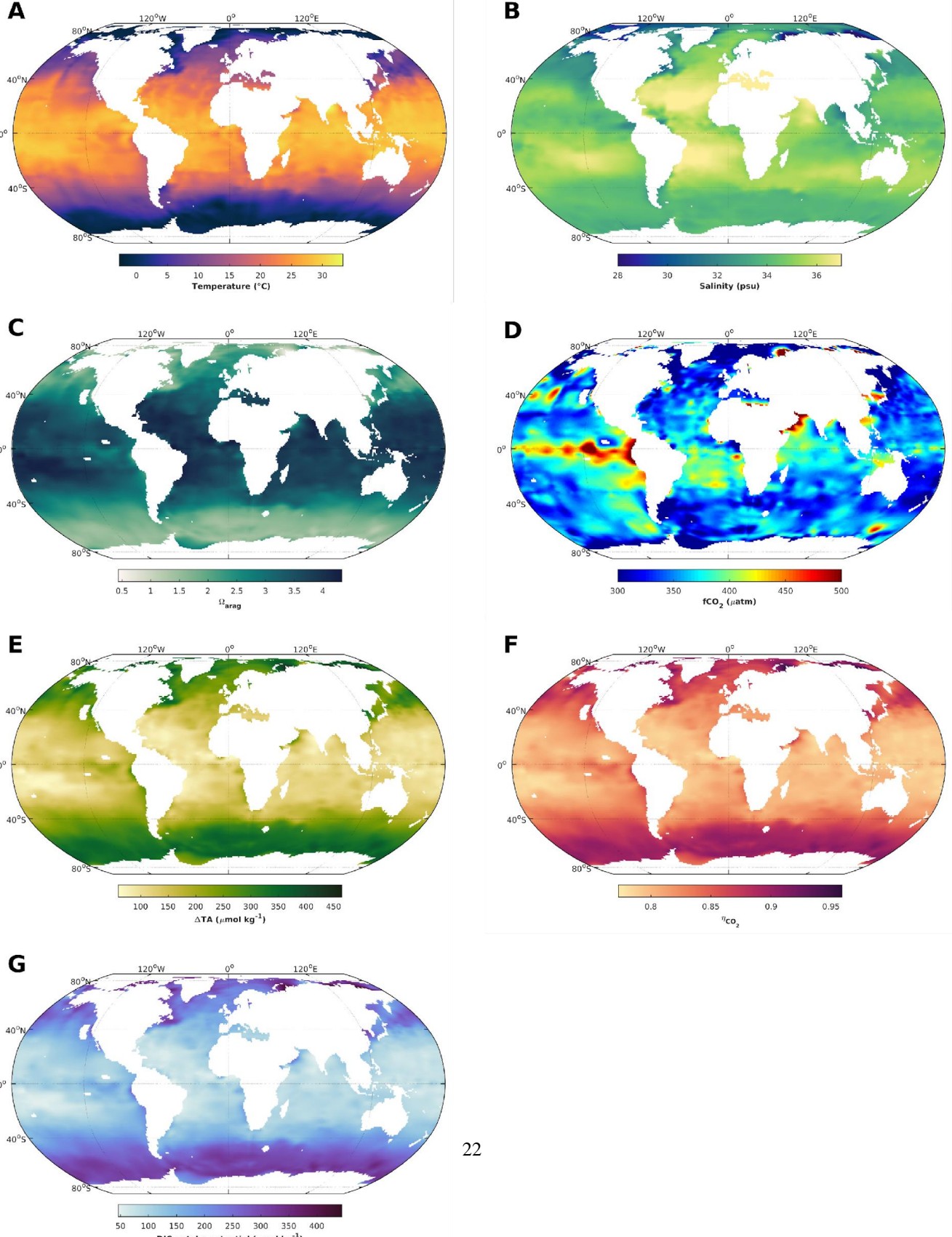

**Figure 2:** 1x1 degree GLODAP climatology (Lauvset et al. 2016) for surface ocean (upper 30 m) temperature (A), salinity (B), aragonite saturation state (C) and $f$CO2 (D), the latter two calculated from GLODAP dissolved inorganic carbon and total alkalinity using the stoichiometric dissociation constants for carbonic acid from Sulpis et al. (2020), for sulphuric acid by Dickson et al. 1990, and total boron from Uppström (1974), the concentration by which total alkalinity can be increased in order to reach an aragonite saturation state of 5, $\Delta$TA (E), the $CO_2$ uptake factor $\eta CO_2$ (F), and the resulting increase in DIC by uptake of atmospheric $CO_2$ upon air-sea equilibration (G). For details see text.

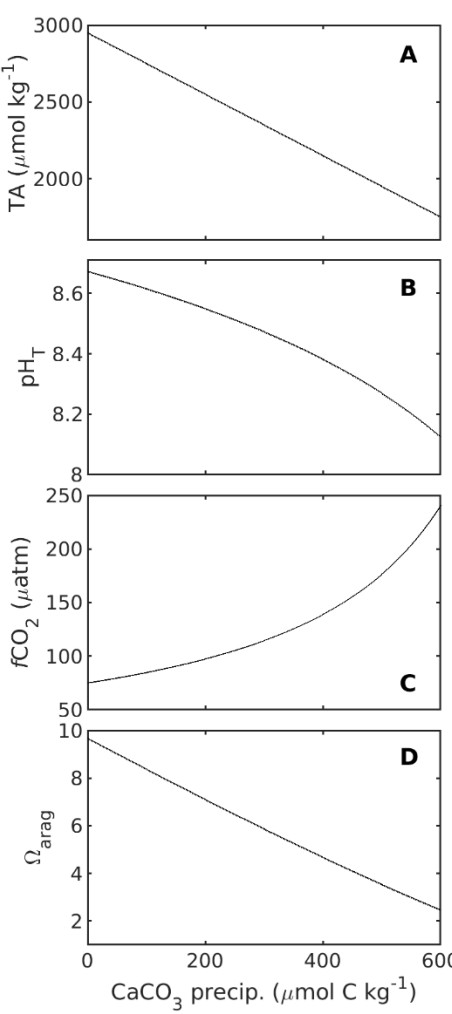

**Figure 3:** Changes in TA (A), $pH_T$ (B), $fCO_2$ (C) and $\Omega_{arag}$ (D) in response to $CaCO_3$ precipitation, i.e. DIC removal. Carbonate chemistry calculations were for a salinity of 35 at 20 °C as described in Figure caption 1. Starting TA and DIC were 2950 and 2100 μmol kg⁻¹. Please note that for 1 mol of $CaCO_3$ precipitated, 1 mol of DIC is consumed.



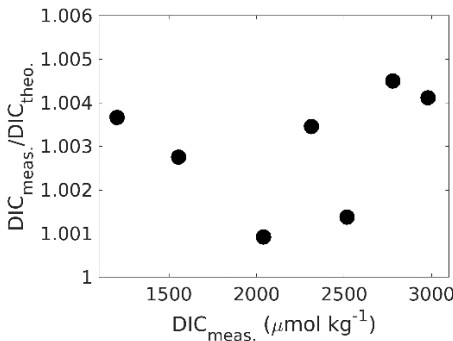

**Figure 4:** Measured DIC concentrations versus the ratio of measured to theoretical DIC, showing that DIC is measured between ~0.1 to 0.4% higher than theoretically predicted from self-prepared $Na_2CO_3$ solutions, without obvious trend across the DIC range. If the instrument response would not be linear, this ratio would change with concentration changes. When linear, the mean ratio of measured to theoretical DIC allows for correcting measured concentrations for any accuracy bias.

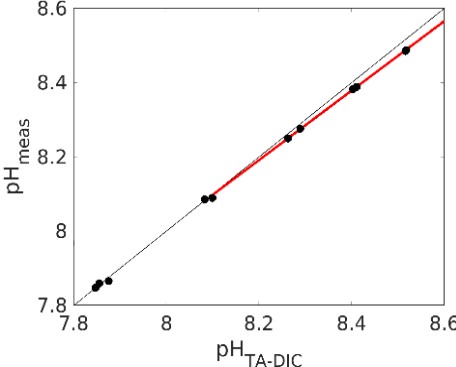

**Figure 5:** Calculated (from measured DIC and TA, see Figure caption 1 for details) vs. measured pH (total scale) using an unpurified batch of mCP. The red line denotes a linear fit through the data above a pH of 8.1.

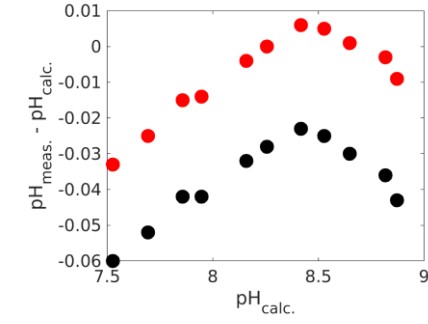


**Figure 6:** Calculated pH (from measured DIC and TA) versus the difference between measured and calculated pH (total scale). The black dots are values prior to the impurity correction and the red ones after (see text for details).