# Peer review of "Seawater carbonate chemistry considerations for ocean alkalinity enhancement research: theory, measurements and calculations"

_State of the Planet, 2023_

## Author Comment (AC1)

Please find replies to the comments of reviewer 1 below in red.

1) This may be a more appropriate comment for the "Climate targets, carbon dioxide removal and the potential role of Ocean Alkalinity Enhancement" chapter (which I assume is the GBP's introduction), but there is inconsistent use of "marine CDR" (*Climate targets...*) vs. the shortened "mCDR" here.

For the OAE guide, we have decided to abbreviate marine carbon dioxide removal with mCDR.

2) Line 41: this should read "with Ksp describing the temperature, salinity AND PRESSURE dependent…"

Thank you for pointing this out. The omission has been corrected.

3) Line 46 and throughout: it is unclear what is meant by "amount contents"

Amount content was defined on line 29 as the amount content of atoms in a seawater sample

4) The description of the Deffeyes diagrams should be made clearer and additional colour should be considered for those figures. The light grey and dark grey lines are difficult to distinguish from the black.

We have modified the Deffeys diagrams, as suggested, to make them more easy to read.

5) Line 315: missing a reference

Thank you for pointing this out, we have added a reference.

6) Line 338: given the earlier focus on saturation state (and its importance to OAE), the brief intro to 'weather' vs 'climate' definitions in this paragraph would benefit from a short sentence clarifying that these definitions are based on achieving a desired uncertainty in [carbonate] (or for 'climate' a *change* in [carbonate])

The 'climate' or 'weather' definition applies to all carbonate chemistry species, whether they are measured directly, e.g. DIC, or calculated, e.g. DIC from measured TA and pH, or calcium carbonate saturation state. It depends what parameter one is interested in.

7) Figure 4: link the caption directly back to how this is used to check for linearity of response.

Thank you for the suggestion, we have modified the caption accordingly.

8) Throughout: some issues with missing italics for "f" in fCO2, and occasionally italicised chemical symbols.

We went through the text and turned the f in fCO2 italic and removed any italic font from chemical symbols.

9) Line 390: delete or clarify "(compare Figure 4)"

We have added the notion 'like for DIC' and also amended the caption for Figure 4.

10) Section 2.3.3: Although the authors are correct to caution (at length) about the use of unpurified dye, someone new to the field may be unaware of the potential downfalls of potentiometric measurements. It would be helpful to include a reference to Bockmon's 2015 "An inter-laboratory comparison assessing the quality of seawater carbon dioxide measurements" paper and highlight how quickly the uncertainty of a potentiometric can exceed the limits for climate/weather measurements.

Thank you for this suggestion, we have added the Bockmon & Dickson 2015 reference.

---

## Author Comment (AC2)

Please find replies to the comments of reviewer 1 below in red.

Section 2.1.1: The paper begins with a useful, if brief, review of the acid-base equilibrium chemistry of the carbonate system in sweater. While these definitions are not new, and the descriptions not exhaustive, I found these sections useful.

Section 2.1.2: I found the description of Figure 1 (Deffeyes diagrams) confusing. The figures themselves would benefit from revision that makes the different sources of alkalinity (NaOH, NaHCO3, etc) more clear (coloured and not grey lines, perhaps)? The subsection would also benefit from a worked example using the figure, in addition to the rather vague statement that it is 'straightforward to assess the potential of 1 kg of a particular modified seawater to remove CO2 from the atmosphere'. Since it is straightforward, please add it, for 1 kg of each of the chosen sources of TA.

We agree and have updated the Deffeys diagrams to include colour. Also, we now explain that assessing the $CO_2$ removal potential involves horizontally moving from the arrow tip that represents a certain alkaline agent addition to the initial $fCO_2$.

Section 2.1.3: while this section is called 'OAE impacts on seawater' it was dominated by descriptions of $CaCO_3$ precipitation, and a disproportionate amount of detail is given to inorganic precipitation relative to the other sections.

We have added the potential for secondary CaCO3 precipitation to the section title to better reflect the content. Some details on the three modes of CaCO3 precipitation were deemed necessary as all three modes will have different saturation state thresholds.

Section 2.1.4: would remove the (strangely casual) 'So, how much TA can be added, then?', and rather refocus on recommendations, though this seems to be a bit out of context with the goals of the chapter (as stated in the abstract).

As suggested, we rephrased the respective sentence. Recommendations are difficult in this section as it is providing a basic understanding on the impacts of OAE on seawater carbonate chemistry and how much of atmospheric $CO_2$ can be taken up.

Section 2.1.4: the focus of this section on the global (or latitudinal) distributions of salinity, temperature and various CO2 system parameters seems rather detailed compared to other sections, including those about sample collection, preservation, and analysis, which were listed as key to the chapter. This doesn't lead to recommendations about where to do the TA additions, but rather ends with a statement about how the uptake factor is minor, and the potential is driven by the amount of added TA (keeping away from critical thresholds in Omega)….

We believe that a key recommendation in this section is that the uptake factor is most likely of minor influence and should not guide decisions where to do OAE and where not.  It is rather the two step assessment by how much alkalinity can be increased without triggering secondary $CaCO_3$ precipitation and how much of $CO_2$ is then actually taken up.

Section 2.2: it is a pity that there are no references given for any of the suggested modifications to the SOPs for sampling, storage and analysis – very nice to the see recommendations for dealing with elevated concentrations.

Unfortunately, experimental OAE is still in its infancy, hence there are no published articles on testing the various modifications suggested here.

The manuscript seems to be missing a (short) conclusion section. In general, while relatively well-written, the text leaves the impression that it was done in a rush, and not with as much care as you might expect from the team of authors.

We agree and have added a paragraph with key recommendations to the end and to the abstract. The guide was indeed on a tight timeline. Nevertheless, we went over the entire manuscript again for refinement.